# Distribution of refractive errors and horizontal concomitant strabismus in Iraqi children

Hassan A. Aljaberi[1,2]*, Saeed Rahmani[2], Humam H. Alrikabi[2]

1 Optical Techniques Department, College of Health and Medical Techniques, Al-Mustaqbal University, Babylon, Iraq, 2 Department of Optometry, Faculty of Rehabilitation, Shahid Beheshti University of Medical Sciences, Tehran, Iran

* hassan.abdulhadi@uomus.edu.iq

## Abstract

Horizontal concomitant strabismus is a common pediatric ocular disorder influenced by underlying refractive error patterns, yet evidence from Iraq remains limited. This study characterized refractive profiles in esotropia and exotropia and identified refractive predictors of each subtype. In this retrospective multicenter clinic-based case-comparison study, medical records of 2,256 children aged 4–15 years evaluated at three tertiary pediatric ophthalmology centers in Iraq were reviewed; 1,012 met the criteria for horizontal concomitant strabismus (642 esotropia and 370 exotropia), while 1,244 non-strabismic children served as controls. All participants underwent standardized cycloplegic refraction, visual acuity assessment, and ocular alignment testing. Group differences were examined using Chi-square tests, and logistic regression was used to identify refractive predictors. Children with esotropia were younger (mean age 8.24±2.89 years) and exhibited a marked hyperopic profile, with 78% having spherical equivalent hyperopia ≥+2.00 D; clinically significant anisometropia (≥2.50 D) was present in 33.8% of cases. Exotropia occurred at a slightly older age (8.75±2.89 years) and was associated with higher proportions of myopia (53.5%) and astigmatism ≥1.00 D (55.7%). Across the full cohort, myopic refractive error (SE<−0.50 D) was present in 310 children (13.7%), hyperopic refractive error (SE≥+2.00 D) in 1,002 (44.4%), astigmatism ≥1.00 D in 555 (24.6%), and anisometropia ≥0.50 D in 1,489 (66.0%). In multivariable analysis, moderate-to-high hyperopia strongly predicted esotropia, whereas myopia and clinically significant astigmatism independently predicted exotropia. These findings show contrasting refractive profiles in Iraqi children with esotropia and exotropia and support the clinical value of early cycloplegic refraction in children with suspected ocular misalignment.

**Data availability statement:** The minimal anonymized dataset underlying the findings of this study is fully available without restriction. All relevant data required to replicate the results are provided within the manuscript and its Supporting Information files.

**Funding:** The author(s) received no specific funding for this work.

**Competing interests:** The authors have declared that no competing interests exist.

## Introduction

Strabismus is one of the most common and clinically significant ocular disorders in childhood [1,2]. It is characterized by a misalignment of the visual axes that disrupts normal binocular vision development and may lead to amblyopia, reduced stereopsis, and long-term visual impairment when not detected early [1–3]. Horizontal concomitant deviations, especially esotropia and exotropia, account for most pediatric strabismus cases and represent an important public health concern due to their impact on visual development, psychosocial functioning, and academic performance [1,3,4]. Understanding the etiological factors contributing to these deviations is essential for improving early detection, guiding individualized management, and reducing the risk of amblyopia [3,4].

Refractive errors are among the most influential contributors to childhood strabismus. The relationship between refractive status and ocular alignment is mediated through accommodation, vergence responses, and the stability of fusional reserves [5]. Hyperopia increases accommodative effort, which may trigger excessive accommodative convergence and predispose children to esotropia [6,7]. Conversely, myopia and reduced accommodative demand have been associated with the development and instability of exotropia, particularly in older children [7,8]. Astigmatism and anisometropia also influence ocular alignment by inducing retinal blur, weakening sensory fusion, and increasing the likelihood of strabismus and amblyopia [9]. Globally, uncorrected refractive error remains a leading cause of visual impairment in children, and its prevalence varies substantially across different regions [10].

Large epidemiological studies from East Asia have provided valuable insights into refractive–strabismus associations. These studies consistently report distinct refractive error patterns and clear dose–response relationships between spherical equivalent refractive error and strabismus subtype [9,11,12]. Higher levels of hyperopia show strong associations with esotropia, whereas myopia and significant astigmatism are more frequently linked to exotropia [9,12]. These findings highlight both universal physiological mechanisms and marked regional differences in refractive development. The high prevalence of myopia in East Asian populations, for instance, shapes refractive profiles that differ noticeably from those observed in Western and Middle Eastern regions [1,9].

In contrast, research from the Middle East and Iraq in particular remains limited. Existing Iraqi studies have primarily focused on describing the prevalence of refractive errors among school-aged children and adolescents, showing substantial burdens of hyperopia, myopia, and astigmatism, as well as their effects on academic and functional performance [13,14]. Additional work has examined ocular biometric characteristics among young adult Iraqis, emphasizing the roles of axial length and corneal curvature in determining refractive status [15,16]. However, these studies have not systematically evaluated how refractive errors relate to specific strabismus subtypes in children, and current evidence is restricted to small, single-center preschool cohorts [17,18].

To date, no large multicenter studies have investigated the association between refractive error patterns and horizontal concomitant strabismus in Iraqi children. This

gap in the literature presents significant challenges for clinical practice. In the absence of robust, population-specific data, clinicians may rely on international findings that do not fully reflect the refractive characteristics, environmental exposures, or developmental profiles of Iraqi children [14–21]. Consequently, opportunities for early detection, targeted screening programs, and informed risk-based counseling may be limited.

A multicenter investigation is therefore needed to address this gap. By assessing refractive error patterns in children diagnosed with esotropia and exotropia and comparing them with those of non-strabismic peers, it is possible to identify associations unique to this population. Such evidence is crucial for informing clinical decision-making, refining referral pathways, strengthening amblyopia prevention strategies, and supporting future research on visual development in Iraq.

This study characterizes the refractive error profiles of Iraqi children with horizontal concomitant strabismus and evaluates the associations between spherical equivalent refractive error, anisometropia, astigmatism, and strabismus subtype. By comparing children with esotropia, exotropia, and non-strabismic controls across multiple pediatric eye centers, this study provides the first large-scale multicenter Iraqi clinic-based assessment of refractive–strabismus relationships in children. It also offers a unified analysis of refractive profiles across clinically relevant subgroups in a setting where comparable pediatric data remain limited.

## Materials and methods

### Study design and clinical setting

This retrospective, multicenter, clinic-based case-comparison study was conducted at three major pediatric ophthalmology centers in Iraq: Ibn Al-Haitham Teaching Eye Hospital (Baghdad), Al-Hakeem Teaching Hospital (Najaf), and Al-Hussain Specialized Eye Complex (Karbala). These high-volume tertiary institutions follow standardized diagnostic protocols and collectively serve a broad urban and semi-urban population. Medical records from December 2023 to June 2025 were reviewed.

Children presenting with suspected ocular deviation or visual complaints were examined consecutively. Those diagnosed with horizontal concomitant strabismus were classified as cases, while non-strabismic children evaluated during the same period served as the comparison group. Accordingly, the proportion of strabismus cases in the analytic sample reflects the clinic-based case-comparison design and should not be interpreted as population prevalence.

### Study population and eligibility criteria

Of the 2,256 children examined (aged 4–15 years), only those whose first documented visit included complete data on ocular alignment, cycloplegic refraction, visual acuity, and anterior- and posterior-segment examinations were eligible for inclusion.

Exclusion criteria included prior strabismus surgery, media opacities affecting alignment assessment, paralytic or restrictive strabismus, vertical deviations, and syndromic or congenital forms such as Down syndrome or Duane retraction syndrome.

After applying these criteria, 1,012 children met the definition of horizontal concomitant strabismus, comprising 642 cases of esotropia and 370 cases of exotropia. The final analytic sample included 647 females and 365 males.

### Visual acuity and ocular examination

Distance visual acuity (VA) was assessed monocularly at 6 meters using Lea Symbols for pre-literate children and ETDRS charts for literate children. When subjective refraction was not feasible, BCVA was estimated from cycloplegic retinoscopy. VA was recorded in Snellen format and converted to logMAR for analysis. Poor visual acuity was defined as logMAR ≥ 0.3.

All children underwent comprehensive anterior-segment and dilated-fundus examinations to exclude structural abnormalities.

## Assessment of strabismus and AC/A ratio

Ocular alignment was evaluated at distance (6 m) and near (33 cm) fixation using the cover–uncover and alternating cover tests. In less cooperative children, the Hirschberg test served as an initial screen. Prism alternate cover testing was used to quantify deviations. Ocular motility assessment in nine gaze positions ensured exclusion of paralytic or restrictive etiologies. Strabismus was classified as esotropia or exotropia and further categorized as constant or intermittent.

When available in the medical records, the AC/A ratio was documented; however, these data were not consistently recorded across centers and were therefore insufficiently complete for formal analysis.

## Cycloplegic refraction and refractive error definitions

Cycloplegia was induced with two drops of 1% cyclopentolate administered five minutes apart, with a third dose given if dilation was inadequate. Cycloplegic retinoscopy or autorefraction was performed following standardized pediatric refraction protocols.

Spherical equivalent (SE) was calculated as sphere + ½ cylinder. Refractive errors were categorized as follows:

- Emmetropia: $-0.50\,D \le SE < +0.50\,D$

- Low myopia: $-3.00\,D \le SE < -0.50\,D$

- Low hyperopia: $+0.50\,D \le SE < +2.00\,D$

- Moderate hyperopia: $+2.00\,D \le SE < +4.00\,D$

- High hyperopia: $+4.00\,D \le SE < +6.00\,D$

Astigmatism was defined as cylindrical refractive error ≥1.00 D and was further classified into simple, compound, and mixed forms according to the relationship of the principal meridians' focal lines to the retina [22,23]. Anisometropia was defined as the absolute interocular difference in spherical-equivalent refraction; for descriptive analyses, a threshold of ≥0.50 D was used, while larger interocular differences were interpreted as indicating greater refractive asymmetry [24,25].

## Eye selection for statistical analysis

To avoid inter-eye correlation, only one eye per child was included in the main regression analyses. In unilateral strabismus, the deviating eye was selected; in alternating deviations, the less hyperopic eye was used. When both eyes were equivalent, the right eye was selected. By contrast, both eyes were used to calculate anisometropia, defined as the interocular difference in spherical equivalent refractive error.

## Statistical analysis

Analyses were performed using IBM SPSS Statistics version 28. Categorical variables were reported as frequencies and percentages, whereas continuous variables were summarized as mean ± standard deviation. Group differences between esotropia and exotropia were evaluated using the Chi-square or Fisher's exact test.

Univariate logistic regression was conducted to identify refractive predictors of strabismus subtype. Variables with $p < 0.10$ in univariate screening were entered into the multivariable model as candidate covariates to avoid premature exclusion of potentially important predictors, whereas final statistical significance in the adjusted as $p < 0.05$. Adjusted odds ratios (ORs) and 95% confidence intervals (CIs) were reported.

## Ethical approval

This study adhered to the principles of the Declaration of Helsinki and received approval from the institutional review boards of all participating centers. Ethical approvals were obtained from the Ibn Al-Haitham Teaching Eye Hospital (IRB/

IAH/2025–112), Imam Hussain Specialized Eye Complex – Faculty of Medicine (IRB/IHC/2025–129), and Al-Hakeem Teaching Hospital (IRB/HTH/2025–099). Because the study involved a retrospective review of de-identified medical records, informed consent was waived by all approving committees.

### Assessment of bias

Selection bias was minimized through consecutive case recruitment and consistent inclusion/exclusion criteria across centers. Information bias was mitigated by standardized examination and refraction protocols. However, limitations inherent to retrospective data collection are acknowledged.

### Sensitivity analysis

Sensitivity analyses were conducted by excluding (1) children with borderline hyperopia (+0.50 to +1.00 D) and (2) cases missing AC/A measurements. The consistency of effect estimates across all models indicated robust findings.

## Results

### Study subjects

A total of 2,256 children aged 4–15 years were examined during the study period. After applying the eligibility criteria, 1,012 children were identified as having horizontal concomitant strabismus, while 1,244 children exhibited no manifest ocular deviation and were included as the comparison group.

Within the strabismic cohort, esotropia was more frequent than exotropia, accounting for 63.4% (642/1,012) and 36.6% (370/1,012), respectively. The mean age was $8.24 \pm 2.89$ years in the esotropia group and $8.75 \pm 2.89$ years in the exotropia group, whereas the non-strabismic group had a mean age of $8.44 \pm 2.90$ years. Female children comprised 64.0% (647/1,012) of the strabismic cohort, whereas males accounted for 36.0% (365/1,012).

Although AC/A ratio was documented when available, the retrospective data were incomplete and heterogeneous across centers, which limited data quality and precluded reliable statistical evaluation. For this reason, AC/A findings were not presented in the Results.

### Demographic and refractive characteristics

Across the full analytic cohort (n = 2,256), myopic refractive error (SE < −0.50 D) was present in 310 children (13.7%), hyperopic refractive error (SE ≥ +2.00 D) in 1,002 children (44.4%), astigmatism ≥1.00 D in 555 children (24.6%), and anisometropia ≥0.50 D in 1,489 children (66.0%). Table 1 presents the distribution of demographic and refractive characteristics across children with esotropia, exotropia, and those without strabismus. Esotropia was most common among children aged 4–6 years, whereas exotropia was more prevalent among older children, particularly those aged 10–12 years. Fig 1 illustrates the age distribution across strabismus categories and reflects the same pattern, with esotropia concentrated in younger ages and exotropia increasing in frequency among older children.

Refractive error patterns also varied substantially between groups. Esotropia demonstrated a pronounced hyperopic shift, with the highest proportions found in the +2.00 to <+4.00 D and +4.00 to <+6.00 D categories. Exotropia, in contrast, was associated with a broader refractive spectrum extending into emmetropia and myopia. Astigmatism ≥1.00 D was most frequent in exotropia, whereas anisometropia ≥1.00 D was more common in esotropia. The severity distribution of astigmatism across esotropia, exotropia, and non-strabismic children is shown in Fig 2, highlighting the higher burden of significant astigmatism in exotropia.

Fig 3 illustrates the distribution of cycloplegic spherical-equivalent refractive error across the three groups. Children with esotropia demonstrate a clearly shifted hyperopic profile, with higher median and narrower interquartile ranges. In contrast, exotropia displays a wider refractive spread extending into emmetropic and myopic values. The non-strabismic group shows an intermediate distribution, predominantly mild hyperopia with a broader overall range.

**Table 1. Distribution of demographic and refractive characteristics across esotropia, exotropia, and non-strabismus groups.**

| Risk factor | Level | Esotropia n (%) | Exotropia n (%) | No strabismus n (%) | P value |
|---|---|---|---|---|---|
| **Age group** | 4–6 y | 217 (33.8) | 101 (27.3) | 395 (31.8) | 0.211 |
| | 7–9 y | 192 (29.9) | 103 (27.8) | 374 (30.1) | |
| | 10–12 y | 176 (27.4) | 125 (33.8) | 355 (28.5) | |
| | 13–15 y | 57 (8.9) | 41 (11.1) | 120 (9.6) | |
| **Sex** | Male | 219 (34.1) | 146 (39.5) | 600 (48.2) | <0.001 |
| | Female | 423 (65.9) | 224 (60.5) | 644 (51.8) | |
| **SE anisometropia (D)** | <0.50 D | 38 (5.9) | 65 (17.6) | 664 (53.4) | <0.001 |
| | 0.50–<1.00 D | 48 (7.5) | 96 (25.9) | 389 (31.3) | |
| | 1.00–<2.50 D | 339 (52.8) | 200 (54.1) | 160 (12.9) | |
| | ≥2.50 D | 217 (33.8) | 9 (2.4) | 31 (2.5) | |
| **SE refractive error (D)** | <−2.00 D | 0 (0.0) | 44 (11.9) | 58 (4.7) | <0.001 |
| | −2.00 to <0.00 D | 6 (0.9) | 154 (41.6) | 105 (8.4) | |
| | 0.00 to <+2.00 D | 107 (16.7) | 130 (35.1) | 650 (52.3) | |
| | +2.00 to <+4.00 D | 393 (61.2) | 40 (10.8) | 342 (27.5) | |
| | +4.00 to <+6.00 D | 134 (20.9) | 2 (0.5) | 85 (6.8) | |
| | ≥+6.00 D | 2 (0.3) | 0 (0.0) | 4 (0.3) | |
| **Astigmatism (D)** | <0.50 D | 228 (35.5) | 80 (21.6) | 676 (54.3) | <0.001 |
| | 0.50–<1.00 D | 260 (40.5) | 84 (22.7) | 373 (30.0) | |
| | ≥1.00 D | 154 (24.0) | 206 (55.7) | 195 (15.7) | |
| **BCVA (logMAR)** | ≤0.3 | 39 (6.1) | 159 (43.0) | 1102 (88.6) | <0.001 |
| | >0.3 | 603 (93.9) | 211 (57.0) | 142 (11.4) | |

The reported p-values represent overall omnibus comparisons across the three study groups. For variables with statistically significant overall differences, Bonferroni-adjusted post hoc pairwise comparisons were conducted to identify specific between-group differences.

### Refractive error distributions across gender and age

Fig 4 displays the distribution of refractive error types by gender. Males and females showed differing refractive profiles, with females demonstrating higher hyperopic proportions and males showing relatively more emmetropia and low myopia.

Fig 5 presents the distribution of refractive error types across age groups, illustrating how hyperopia, myopia, and emmetropia vary through childhood.

### Associations between refractive error and strabismus subtype

Associations between refractive parameters and esotropia are shown in Table 2, which includes univariate and multi-variable logistic regression models. Hyperopia emerged as the strongest predictor of esotropia, displaying a clear dose–response relationship. Clinically significant anisometropia (≥1.00 D) also remained an independent risk factor following adjustment for demographic and refractive variables.

Corresponding associations for exotropia are shown in Table 3. Myopic refractive error (SE <−0.50 D) was the most influential predictor of exotropia and remained strongly significant in adjusted models. Astigmatism ≥1.00 D also remained an independent risk factor, whereas increasing hyperopia showed a protective effect.

## Discussion

This multicenter clinic-based case-comparison study provides the first large-scale Iraqi analysis of refractive profiles associated with horizontal concomitant strabismus in children and adds region-specific evidence from a setting in which

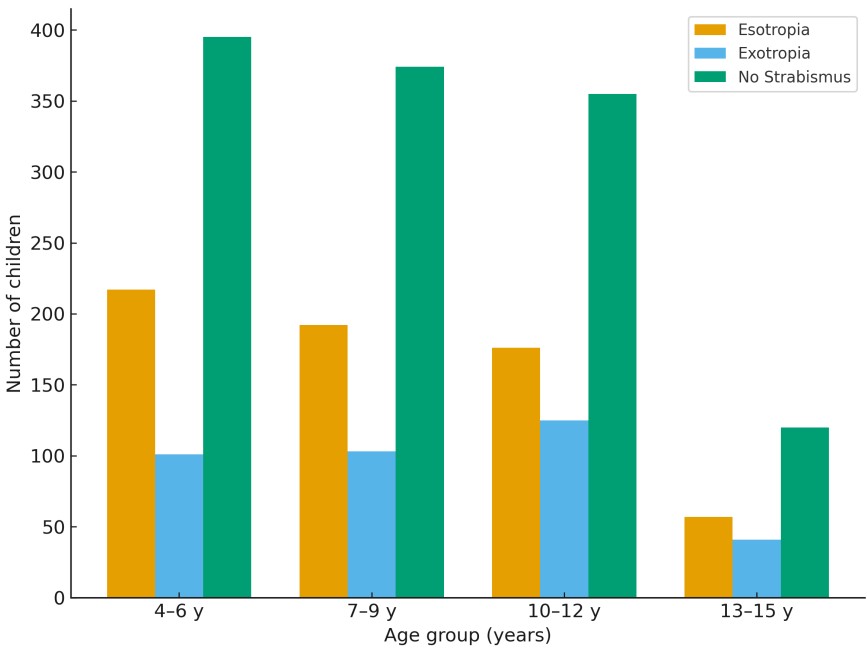

**Fig 1. Age distribution across strabismus categories.** No statistically significant overall difference was observed among the groups (p = 0.211).

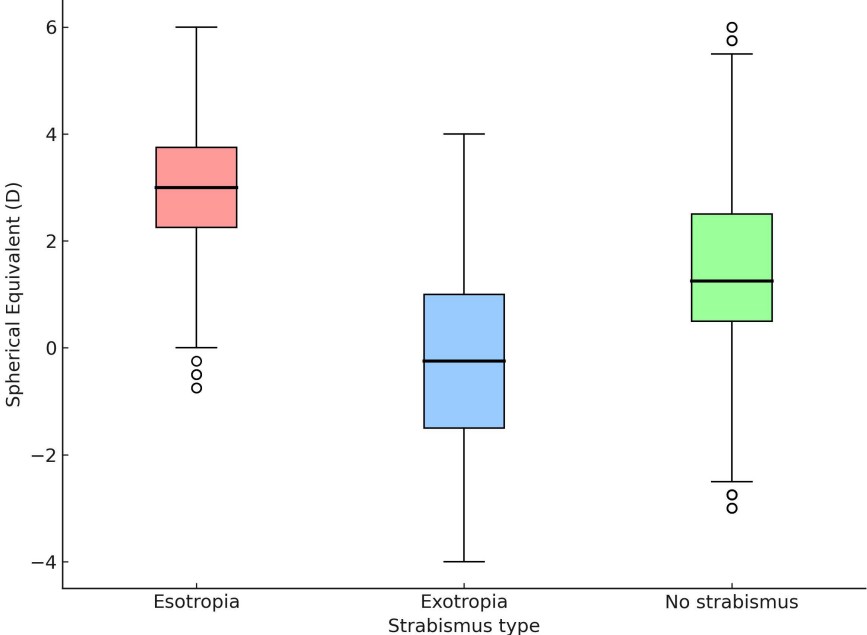

**Fig 2. Prevalence of astigmatism levels across strabismus subtypes.** Overall comparison across groups was statistically significant (p < 0.001).

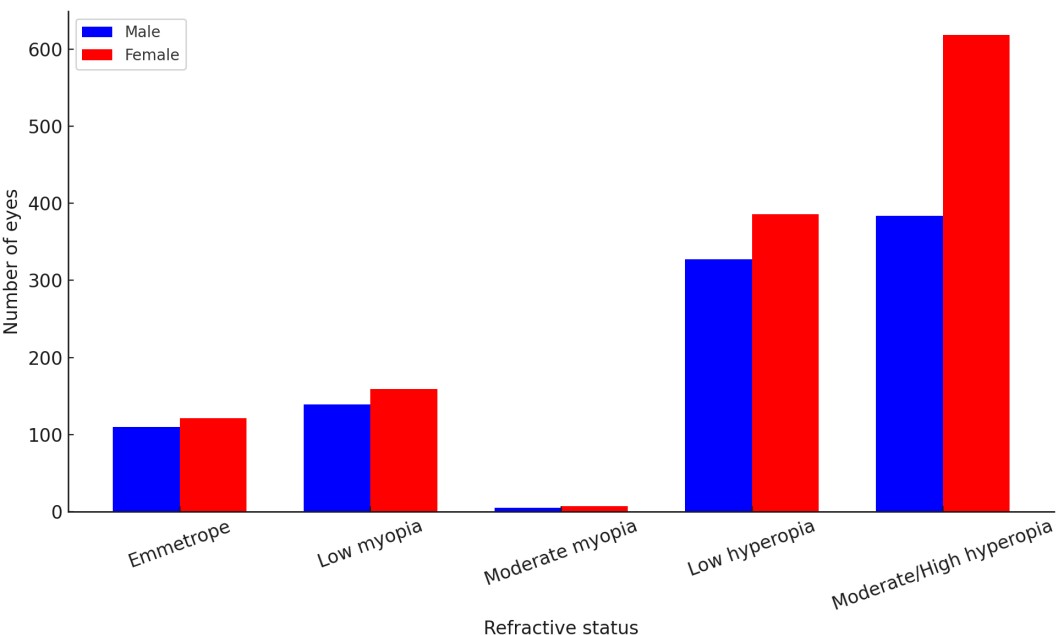

**Fig 3. Boxplot of spherical equivalent refractive error by strabismus type.** Overall omnibus comparison across the three groups was statistically significant (p<0.001). The boxplots show the median, interquartile range, and range; circles indicate outliers.

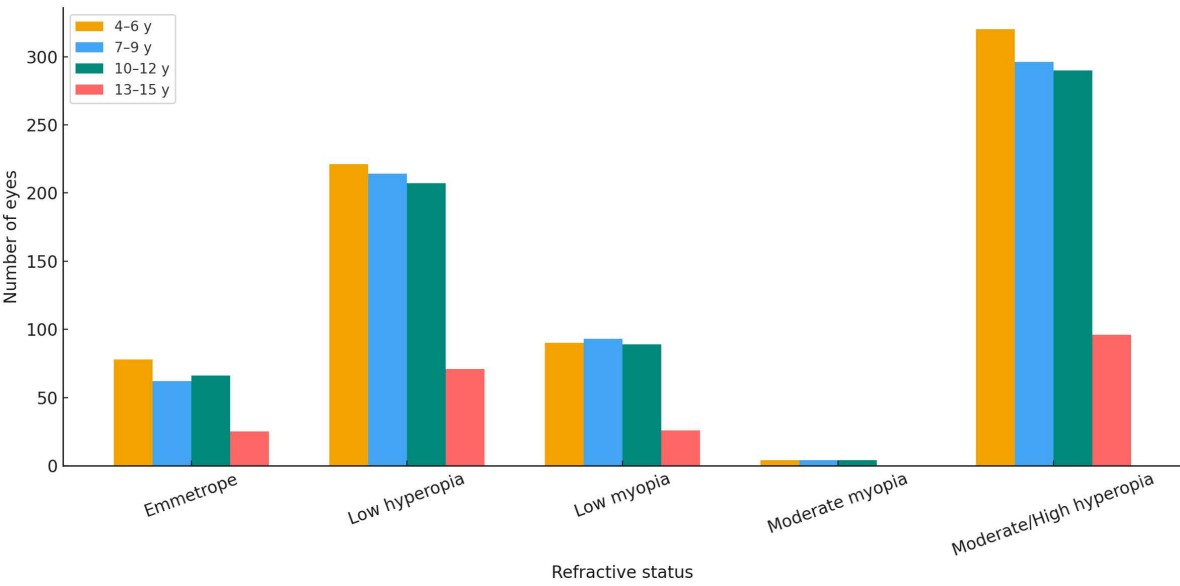

**Fig 4. Distribution of refractive error types by gender.** The overall association between sex and refractive status was statistically significant (p<0.001). Values are presented as number of eyes.

comparable pediatric data remain limited. Although the overall refractive–strabismus patterns observed in the present cohort are broadly consistent with the international literature, this study contributes clinically relevant local data by quantifying the relative distributions of hyperopia, myopia, astigmatism, anisometropia, age, sex, and visual acuity across

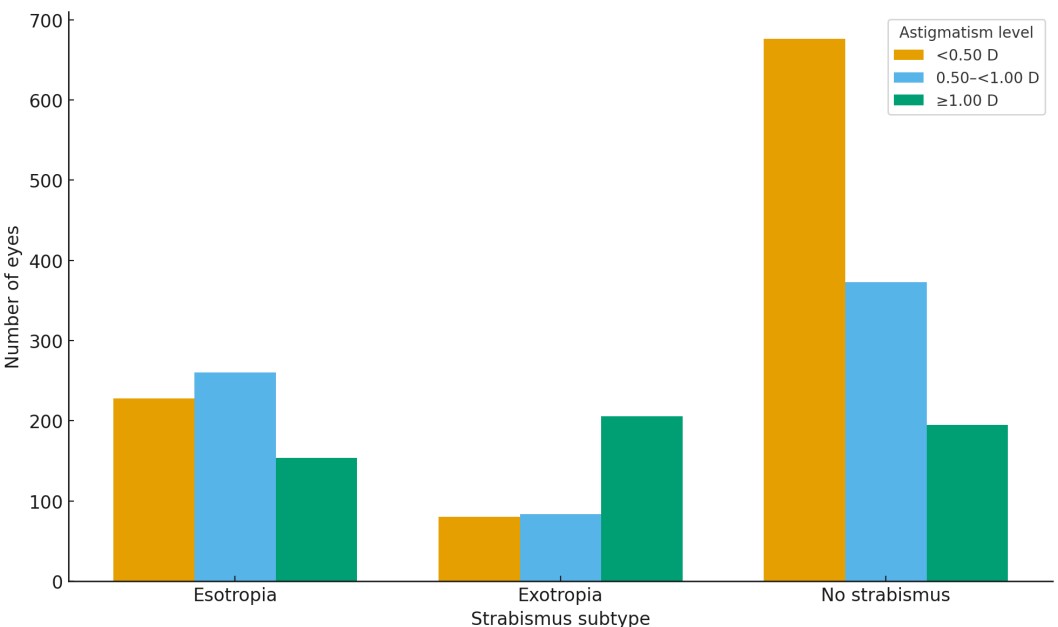

**Fig 5. Distribution of refractive error types across age groups.** Overall association between age group and refractive status was statistically significant (p<0.001). Values represent the number of eyes.

**Table 2. Univariate and multivariate analyses for associations between refractive error and esotropia.**

| Factor | Level | Univariate | | | Multivariate | | |
|---|---|---|---|---|---|---|---|
| | | OR | 95% CI | p | OR | 95% CI | p |
| **Age group** | 4–6 y (Ref) | 1.00 | – | – | 1.00 | – | – |
| | 7–9 y | 0.92 | 0.73–1.16 | 0.480 | 1.01 | 0.70–1.44 | 0.976 |
| | 10–12 y | 0.84 | 0.66–1.06 | 0.141 | 1.00 | 0.70–1.42 | 0.995 |
| | 13–15 y | 0.81 | 0.58–1.14 | 0.225 | 1.04 | 0.63–1.70 | 0.882 |
| **Sex** | Male (Ref) | 1.00 | – | – | 1.00 | – | – |
| | Female | 1.66 | 1.37–2.01 | <0.001 | 1.57 | 1.18–2.09 | 0.002 |
| **SE anisometropia (D)** | <0.50 (Ref) | 1.00 | – | – | 1.00 | – | – |
| | 0.50 to <1.00 D | 1.90 | 1.22–2.95 | 0.004 | 2.17 | 1.36–3.47 | 0.001 |
| | 1.00 to <2.50 D | 18.07 | 12.63–25.85 | <0.001 | 26.43 | 17.65–39.56 | <0.001 |
| | ≥2.50 D | 104.07 | 65.10–166.38 | <0.001 | 114.78 | 67.04–196.51 | <0.001 |
| **SE refractive error (D)** | 0.00 to <+2.00 D (Ref) | 1.00 | – | – | 1.00 | – | – |
| | <0.00 D | 0.12 | 0.05–0.28 | <0.001 | 0.08 | 0.04–0.20 | <0.001 |
| | +2.00 to <+4.00 D | 7.50 | 5.86–9.59 | <0.001 | 8.12 | 5.90–11.19 | <0.001 |
| | ≥+4.00 D | 10.89 | 7.80–15.21 | <0.001 | 14.63 | 9.22–23.21 | <0.001 |
| **Astigmatism (D)** | 0.00–<0.50 D (Ref) | 1.00 | – | – | 1.00 | – | – |
| | 0.50 to <1.00 D | 1.89 | 1.53–2.33 | <0.001 | 1.98 | 1.44–2.73 | <0.001 |
| | ≥1.00 D | 1.27 | 1.00–1.61 | 0.046 | 1.48 | 1.03–2.12 | 0.032 |
| **BCVA (logMAR)** | ≤0.3 (Ref) | 1.00 | – | – | 1.00 | – | – |
| | >0.3 | 55.23 | 39.13–77.96 | <0.001 | 54.17 | 38.36–76.48 | <0.001 |

**Table 3. Univariate and multivariate analyses for associations between refractive error and exotropia.**

| Factor | Level | Univariate | | | Multivariate | | |
|---|---|---|---|---|---|---|---|
| | | OR | 95% CI | p | OR | 95% CI | p |
| **Age group** | 4–6 y (Ref) | 1.00 | – | – | 1.00 | – | – |
| | 7–9 y | 1.10 | 0.82–1.48 | 0.519 | 1.17 | 0.80–1.70 | 0.414 |
| | 10–12 y | 1.43 | 1.07–1.90 | 0.015 | 1.48 | 1.03–2.13 | 0.034 |
| | 13–15 y | 1.40 | 0.94–2.09 | 0.096 | 1.46 | 0.87–2.42 | 0.149 |
| **Sex** | Male (Ref) | 1.00 | – | – | 1.00 | – | – |
| | Female | 1.18 | 0.94–1.48 | 0.159 | 1.48 | 1.11–1.98 | 0.008 |
| **SE anisometropia (D)** | <0.50 D (Ref) | 1.00 | – | – | 1.00 | – | – |
| | 0.50–<1.00 D | 2.37 | 1.69–3.32 | <0.001 | 2.64 | 1.78–3.91 | <0.001 |
| | 1.00–<2.50 D | 4.33 | 3.20–5.86 | <0.001 | 7.12 | 4.89–10.37 | <0.001 |
| | ≥2.50 D | 0.39 | 0.19–0.80 | 0.010 | 0.74 | 0.33–1.65 | 0.457 |
| **SE refractive error (D)** | 0.00–<+2.00 D (Ref) | 1.00 | – | – | 1.00 | – | – |
| | <0.00 D | 6.82 | 5.17–9.00 | <0.001 | 6.57 | 4.76–9.08 | <0.001 |
| | +2.00–<+4.00 D | 0.32 | 0.22–0.46 | <0.001 | 0.22 | 0.14–0.33 | <0.001 |
| | ≥+4.00 D | 0.05 | 0.01–0.21 | <0.001 | 0.04 | 0.01–0.15 | <0.001 |
| **Astigmatism (D)** | <0.50 D (Ref) | 1.00 | – | – | 1.00 | – | – |
| | 0.50–<1.00 D | 1.50 | 1.09–2.07 | 0.014 | 1.54 | 1.06–2.22 | 0.023 |
| | ≥1.00 D | 6.67 | 5.01–8.88 | <0.001 | 6.33 | 4.49–8.91 | <0.001 |
| **BCVA (logMAR)** | ≤0.3 (Ref) | 1.00 | – | – | 1.00 | – | – |
| | >0.3 | 2.03 | 1.62–2.55 | <0.001 | 2.01 | 1.60–2.53 | <0.001 |

esotropia, exotropia, and non-strabismic children within a single analytic framework. In this respect, the present findings extend earlier Iraqi reports, which were generally smaller, single-center, or primarily descriptive, and provide a more clinically applicable refractive risk profile for pediatric ophthalmic practice in Iraq [14,18,26,27].

A principal finding was the strong association between esotropia and moderate-to-high hyperopia. Most children with esotropia fell within the +2.00 to <+4.00 D range, and a substantial proportion also had higher hyperopic levels. This pattern corresponds with the classical accommodative convergence mechanism, whereby increased accommodative effort in uncorrected or under-corrected hyperopic eyes may stimulate excessive convergence and precipitate esodeviation [28–30]. At the regional level, the same directional relationship has been described in Iraqi studies and in pediatric cohorts from Iran, Turkey, Jordan, and Saudi Arabia, although the extent of directly comparable evidence varies across countries and across specific refractive subtypes. Overall, these findings support the interpretation that hyperopia remains the dominant refractive correlate of childhood esotropia in this regional context [18,26,29–37].

By contrast, exotropia in the present cohort showed a broader refractive distribution extending toward emmetropia and myopia, with myopia emerging as the strongest refractive predictor in multivariable analysis. This finding is also biologically coherent. Reduced accommodative demand in myopic eyes may lessen accommodative convergence, thereby favoring exodeviation or reducing alignment stability in predisposed children [11,28,32]. The consistency of this pattern with reports from neighboring countries suggests that the myopia–exotropia relationship is not restricted to East Asian populations, even though the absolute burden of childhood myopia may differ across regions [31,33,34,37–40]. From a clinical perspective, these data indicate that exotropia in Iraqi children should not be viewed only through the lens of intermittent deviation or apparent emmetropia, but also in relation to measurable refractive risk, particularly myopic shift.

Astigmatism was another major distinguishing feature of exotropia in this cohort. Astigmatism of at least 1.00 D was substantially more frequent in exotropic than in esotropic and non-strabismic children, and it remained independently associated with exotropia after adjustment. This finding is compatible with the concept that persistent meridional blur

may interfere with binocular sensory development and fusion, particularly when combined with other refractive or developmental vulnerabilities [41,42]. Regional studies likewise suggest that clinically meaningful astigmatism is commonly encountered in pediatric strabismus, even if not all reports distinguish exotropia from esotropia using identical methods [32–34,43]. Rather than treating astigmatism as a secondary refractive finding, the present data support its inclusion in risk-oriented refractive assessment in children presenting with exodeviation.

Anisometropia was particularly prominent in esotropia and showed a strong graded association with strabismus risk. This observation is clinically important because interocular refractive asymmetry may impair binocular sensory development, weaken fusion, and increase vulnerability to amblyopia and manifest deviation [22,43]. In the current cohort, anisometropia was markedly more common in esotropia than in non-strabismic children, and the adjusted models demonstrated a clear dose-response pattern. These findings are consistent with prior work linking anisometropia to accommodative esotropia and broader binocular dysfunction [44,45]. They also reinforce the practical importance of bilateral cycloplegic refraction, because anisometropia can only be adequately characterized through direct comparison of refractive error between fellow eyes [44].

Age-related patterns also warrant interpretation beyond simple description. Esotropia was relatively more concentrated in younger children, whereas exotropia was more frequent in later childhood, particularly around 10–12 years. This age gradient is compatible with the natural course of refractive development, in which younger children more commonly retain hyperopic refractive profiles, whereas later childhood is associated with progressive emmetropization and, in some populations, increasing myopia prevalence [33–35,38,40]. The observed age distribution therefore supports the view that refractive maturation contributes meaningfully to the changing balance between esotropic and exotropic phenotypes during childhood. From a service-delivery perspective, these findings suggest that early preschool surveillance may be particularly relevant for hyperopia-associated esotropia, whereas continued monitoring during school age may be important for myopia- and astigmatism-associated exotropia.

Sex-related differences in the present cohort merit more careful interpretation than a purely descriptive account would allow. Females were more frequent in the strabismic cohort and remained significantly associated with both esotropia and exotropia after multivariable adjustment. This finding should not be interpreted as evidence of a single sex-specific biological mechanism. Rather, it may reflect the combined influence of several factors, including possible sex-related differences in ocular growth patterns, variation in parental concern regarding visible ocular deviation, healthcare-seeking behavior, referral pathways to tertiary centers, and sampling characteristics inherent to clinic-based studies. The regional literature does not show entirely uniform sex-related patterns, which further suggests that the observed female predominance may be context-dependent rather than universally causal [35,38]. Accordingly, the female excess identified in the present cohort is best interpreted as an important epidemiologic feature of the sampled clinical population that warrants further prospective investigation, rather than as definitive evidence of sex-based susceptibility.

The present findings also have practical implications for pediatric ophthalmic care in Iraq. Because the non-strabismic comparison group was predominantly clustered within the mild hyperopic or near-emmetropic range, whereas esotropia was strongly associated with hyperopia of +2.00 D or greater and exotropia was more closely linked to myopia and clinically relevant astigmatism, the results underscore the clinical value of structured cycloplegic refraction as an early risk-stratification tool in children with suspected ocular misalignment. In particular, moderate-to-high hyperopia, meaningful anisometropia, and astigmatism of at least 1.00 D appear to justify closer assessment of ocular alignment and amblyopia-oriented follow-up. However, because this was a clinic-based case-comparison study rather than a population survey, the observed proportions should be interpreted as distributions within the analytic sample rather than as prevalence estimates for Iraqi children as a whole.

This study has several strengths, including its relatively large sample size, multicenter recruitment, and standardized use of cycloplegic refraction. These features strengthen the clinical relevance and internal consistency of the findings. Nevertheless, several limitations should be acknowledged. The retrospective case-comparison design precludes causal

inference and limits generalizability beyond similar tertiary clinical settings. Because the study was conducted in referral pediatric ophthalmology centers, the sample may have overrepresented children with strabismus and clinically significant refractive abnormalities relative to the general pediatric population; therefore, the observed distributions should be interpreted as clinic-based associations rather than population prevalence estimates. Biometric measurements such as axial length and keratometry were unavailable, limiting exploration of the structural determinants of the observed refractive–strabismus associations. Spectacle-correction status at presentation was also not consistently documented, which restricts interpretation of the extent to which prior optical correction may have influenced accommodative demand and ocular alignment. In addition, further subclassification of horizontal strabismus by convergence status and AC/A ratio would have provided clinically relevant phenotypic detail, but these variables were not recorded with sufficient completeness and consistency across centers to support reliable subgroup analysis; accordingly, they were not presented in the Results. Future prospective multicenter studies incorporating biometric parameters, systematically documented correction status, and complete AC/A assessment would allow more precise phenotypic characterization and deeper insight into the mechanisms linking refractive error to childhood horizontal concomitant strabismus.

## Conclusion

This study demonstrates distinct and clinically meaningful differences in the refractive profiles of esotropia and exotropia among Iraqi children. Moderate to high hyperopia and anisometropia strongly characterized esotropia, whereas myopia and significant astigmatism were more commonly associated with exotropia. These patterns closely mirror regional and global evidence and highlight the central role of cycloplegic refraction in the early identification of children at risk for strabismus. The age-related shift from hyperopia in early childhood to increased myopia in later years further emphasizes the need for age-appropriate screening strategies. Strengthening nationwide pediatric vision screening programs, ensuring early optical correction, and raising awareness among parents and primary-care providers may substantially reduce the burden of strabismus and its sequelae, including amblyopia. The findings of this study provide an important epidemiologic foundation for future prospective research and may guide public health planning and clinical decision-making in Iraq and similar settings.

## Supporting information

**S1 Dataset. Anonymized dataset used for the statistical analyses in this study.**
(XLSX)

## Acknowledgments

We would like to express our appreciation to the administrations of the participating pediatric ophthalmology centers for their support in facilitating access to clinical records. We also thank the clinical teams and medical record staff for their assistance with data retrieval, verification, and coordination throughout the study.

## Author contributions

**Conceptualization:** Hassan A. Aljaberi, Saeed Rahmani.

**Data curation:** Hassan A. Aljaberi.

**Formal analysis:** Saeed Rahmani.

**Investigation:** Humam H. Alrikabi.

**Methodology:** Saeed Rahmani.

**Project administration:** Humam H. Alrikabi.

**Validation:** Humam H. Alrikabi.

**Visualization:** Hassan A. Aljaberi, Saeed Rahmani.

**Writing – original draft:** Hassan A. Aljaberi.

**Writing – review & editing:** Saeed Rahmani.

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
