## [Decision Letter · Decision Letter 0]

29 Mar 2026

PONE-D-25-63958Distribution of refractive errors and horizontal concomitant strabismus in Iraqi childrenPLOS One

Dear Dr. Aljaberi,

Thank you for submitting your manuscript to PLOS ONE. After careful consideration, we feel that it has merit but does not fully meet PLOS ONE’s publication criteria as it currently stands. Therefore, we invite you to submit a revised version of the manuscript that addresses the points raised during the review process.

The manuscript was reviewed by two independent reviewers who felt that the paper was well written. However, they requested more emphasis on the unique contribution of the paper. They pointed out that several methodological aspects should be clarified, some results should be presented that were lacking, and that more depth was necessary in the discussion, including the limitations section. I have also reviewed the paper and provided some aspects to incorporate in your revision.

We look forward to receiving your revised manuscript.

Kind regards,

Liat Gantz, PhD

Academic Editor

PLOS One

Journal Requirements:

3. We note that your Data Availability Statement is currently as follows: “All relevant data are within the manuscript and its Supporting Information files.”

4. Please amend your manuscript to include your abstract after the title page.

5. Please remove your figures from within your manuscript file, leaving only the individual TIFF/EPS image files, uploaded separately. These will be automatically included in the reviewers’ PDF.

Additional Editor Comments:

The manuscript has been reviewed by two independent reviewers who felt that the paper was well written. They felt that the unique contribution of the paper should be better stressed, and that several methodological aspects should be clarified. Both felt that some results should be presented that were lacking, and that more depth was necessary in the discussion, including the limitations section.

Additional comments that should be addressed as well, can be seen below:

Uncorrected hyperopia increases accommodative effort

This study aims to characterize? Or characterizes?

A case-comparison design rather than a population based survey

Provide references for the definitions of astigmatism, anisometropia

What does this mean: Only the eyes were used to compute anisometropia? As opposed to the ears?

Statistical analysis: Group differences between esotropia and exotropia were compared instead of evaluated

Why were variables with p-values < 0.10 included in the logistic regression as opposed to p<0.05?

Results Study Subjects

Female/male ratio of only the strabismus group provided, please provide for the non strabismic group as well

Table 1- what do the p-values represent, specify in the legend. The legend and the table should be independently coherent without the full manuscript. Also- after obtaining a significant p-value, since the comparison is between three groups, there should be post-hoc tests with results specified in the table

Figures 1-5 should specify the significant differences using asterisk *

Myopic refractive error was defined as refractive error < 0.50 in the methods but in the results in the “Associations between refractive error and strabismus type”, and in Tables 2-3, it is defined as anything lower than plano. The definition should be consistent throughout.

The sentence in the discussion “The current findings align strongly with regional and global evidence” is missing references

Were the majority of the children uncorrected? If they were corrected then they would not be expected to increase accommodation in hyperopia as explained in the discussion…

In the discussion regarding hyperopia and esotropia you mention Iran, Turkey, and Jordan, without Saudi Arabia

In the discussion on exotropia and emmetropia/myopia you mention Iran, Turkey and Saudi Arabia, without Jordan.

In the discussion on astigatism and exotropia, only Turkey Iran and Jordan are mentioned.

In the discussion regarding females and hyoperopia only Iran and Saudi Arabia are mentioned.

The countries should be consistent in all instances with special attention to cases of mismatch including possible reasoning for mismatches.

“Delayed access to eye examinations, insufficient screening programs, low rates of early optical correction”- missing references. Also appears twice in two different paragraphs in the discussion, omit the redundancy

I would have liked to see the prevalence of the refractive errors in the cohort, this puts the issue of screening in context. For example, it is warranted in the sentence “Early cycloploegic refraction is essential for identifying children at risk of strabismus. Hyperopia > +2.00 D should prompt…”

The discussion limitations state the lack of AC/A although these were included in the methods and nothing was shown in the results. Please explain

Reviewers' comments:

Reviewer's Responses to Questions

**Comments to the Author**

1. Is the manuscript technically sound, and do the data support the conclusions?

Reviewer #1: Yes

Reviewer #2: Yes

2. Has the statistical analysis been performed appropriately and rigorously? 

Reviewer #1: Yes

Reviewer #2: Yes

3. Have the authors made all data underlying the findings in their manuscript fully available?

Reviewer #1: Yes

Reviewer #2: Yes

4. Is the manuscript presented in an intelligible fashion and written in standard English?

Reviewer #1: Yes

Reviewer #2: Yes

5. Review Comments to the Author

Reviewer #1: This multicenter study provides a valuable large-scale assessment of refractive profiles in Iraqi children with horizontal concomitant strabismus. The sample size of 2,256 children is commendable, and the overall methodology appears clinically sound. However, several issues should be addressed to strengthen the manuscript:

1.

Although the dataset is population-specific, the findings closely mirror those reported in international cohorts. The authors should clarify the unique contributions of this study and articulate how these results advance current understanding beyond confirming existing trends.

2.

The Discussion section is primarily descriptive, repeating the Results without offering a sufficiently critical interpretation. For instance, while a high proportion of females (64%) was noted in the strabismus cohort, the potential reasons for this gender distribution are only briefly mentioned and require deeper exploration.

3.

Horizontal strabismus is a complex entity that should be further categorized by convergence status and the AC/A ratio. Although the authors mention measuring the AC/A ratio "when available," it was excluded from the main analysis. Including these data would significantly enhance the clinical depth of the study and help identify specific refractive-strabismus phenotypes within this population.

4.

Please ensure all mathematical symbols and units are consistently formatted throughout the text.

Reviewer #2: Thank you for inviting me to review this manuscript. Overall, this is a well-written paper with adequate sample size. Please find some minor comments to be addressed:

In the study design and clinical setting, you used the word "accessed" instead of "assessed"

"The study followed a case-comparison design, not a population-based survey"...I think this statement isn't needed as you have already stated it's clinic-based. Hence, I suggest you put it in a form like "This retrospective multicenter clinic-based case-comparison study was conducted...".

The clinic-based design might have influenced the oversampling of strabismus cases, hence this should be addressed in the limitations as it's one of the key limitations in this study.

6. PLOS authors have the option to publish the peer review history of their article (what does this mean?). If published, this will include your full peer review and any attached files.

Reviewer #1: No

Reviewer #2: No

---

## [Author Response · Author response to Decision Letter 1]

6 Apr 2026

Please find attached our detailed responses to Reviewer 1 and Reviewer 2, as well as our response to the Editor. We have carefully considered all comments and revised the manuscript accordingly.

---

## [Decision Letter · Decision Letter 1]

5 May 2026

Distribution of refractive errors and horizontal concomitant strabismus in Iraqi children

PONE-D-25-63958R1

Dear Dr. Aljaberi,

We’re pleased to inform you that your manuscript has been judged scientifically suitable for publication and will be formally accepted for publication once it meets all outstanding technical requirements.

Kind regards,

Liat Gantz, PhD

Academic Editor

PLOS One

Additional Editor Comments (optional):

The authors have adequately addressed the reviewers' comments and both have recommended that the article be accepted for publication.

Reviewers' comments:

Reviewer's Responses to Questions

**Comments to the Author**

1. If the authors have adequately addressed your comments raised in a previous round of review and you feel that this manuscript is now acceptable for publication, you may indicate that here to bypass the “Comments to the Author” section, enter your conflict of interest statement in the “Confidential to Editor” section, and submit your "Accept" recommendation.

Reviewer #1: All comments have been addressed

Reviewer #2: All comments have been addressed

2. Is the manuscript technically sound, and do the data support the conclusions?

Reviewer #1: Yes

Reviewer #2: Yes

3. Has the statistical analysis been performed appropriately and rigorously? 

Reviewer #1: Yes

Reviewer #2: Yes

4. Have the authors made all data underlying the findings in their manuscript fully available?

Reviewer #1: Yes

Reviewer #2: Yes

5. Is the manuscript presented in an intelligible fashion and written in standard English?

Reviewer #1: Yes

Reviewer #2: Yes

6. Review Comments to the Author

Reviewer #1: This multicenter study provides a valuable large-scale assessment of refractive profiles in Iraqi children with horizontal concomitant strabismus. The sample size of 2,256 children is commendable, and the overall methodology appears clinically sound. The authors have addressed the concerns.

Reviewer #2: (No Response)

7. PLOS authors have the option to publish the peer review history of their article (what does this mean?). If published, this will include your full peer review and any attached files.

Reviewer #1: No

Reviewer #2: No

---

## [Editor Report · Acceptance letter]

PONE-D-25-63958R1

PLOS One

Dear Dr. Aljaberi,

I'm pleased to inform you that your manuscript has been deemed suitable for publication in PLOS One. Congratulations! Your manuscript is now being handed over to our production team.

Kind regards,

on behalf of

Dr. Liat Gantz

Academic Editor

PLOS One